# Linguini: A benchmark for language-agnostic linguistic reasoning

## Abstract

We propose a new benchmark to measure a language model's linguistic reasoning skills without relying on pre-existing language-specific knowledge. The test covers 894 questions grouped in 160 problems across 75 (mostly) extremely low-resource languages, extracted from the International Linguistic Olympiad corpus. To attain high accuracy on this benchmark, models don't need previous knowledge of the tested language, since all the information required to solve the linguistic puzzle is provided within the context. We find that, while all analyzed models rank below 25% accuracy, there is a significant gap between open and closed models, with the best-performing proprietary model at 24.05% and the best-performing open model at 8.84%.

## 1 Introduction

Recently, language models have shown impressive multilingual skills (Xu et al., 2024), achieving state of the art results in several tasks, such as machine translation (OpenAI, 2024), bilingual lexicon induction (Brown et al., 2020) and cross-lingual classification (Xue et al., 2021). However, the steep performance increase in these tasks has led to the saturation of popular benchmarks, such as MMLU (Hendrycks et al., 2021), where state-of-the-art (SotA) performance has gone from 60% in December 2021 (Rae et al., 2022) to 90% in December 2023 (Gemini Team, 2024), providing diminishing returns when it comes to quantifying differences between models.

Moreover, in the case of linguistic reasoning, the task of evaluating a model's linguistic skills is often tied to the comprehensive knowledge a model has of a certain language (most commonly, English), making it difficult to evaluate a model's underlying linguistic skills beyond language-specific knowledge.

To address these issues, we introduce Linguini[1], a linguistic reasoning benchmark. Linguini consists of linguistic problems which require meta-linguistic awareness and deductive reasoning capabilities to be solved instead of pre-existing language proficiency. Linguini is based on problems extracted from the International Linguistic Olympiad (IOL)[2], a secondary school level contest where participants compete in solving Rosetta Stone-style problems (Derzhanski and Payne, 2010) relying solely on their understanding of linguistic concepts. An example of the type of challenges and the reasoning steps needs to solve it can be seen in Figure 2.

We evaluate a list of open and proprietary models on Linguini, showing a noticeable gap between open and closed language models, in favor of the latter. We also conduct a series of experiments aiming at understanding the role of the contextual information in the accuracy obtained in the benchmark, performing both form

---

[1] The dataset is available at `https://github.com/<redacted>`

[2] The problems are shared only for research purposes under the license CC-BY-SA 4.0. The problems are copyrighted by ©2003-2024 International Linguistic Olympiad

(transliteration) and content (removing context) ablations, with results showing a main reliance on the context to solve the problems, minimizing the impact of language or task contamination in the models' training sets.

## 2 RELATED WORK

There has been an increasing number of articles focusing on evaluating reasoning in language models (Chang et al., 2024). In the area of mathematical reasoning, Qin et al. (2023) analyze models' arithmetic reasoning, while Frieder et al. (2023) leverage publicly-available problems to build GHOSTS, a comprehensive mathematical benchmark in natural language. Bang et al. (2023) include symbolic reasoning in their multitask, multilingual and multimodal evaluation suite. Wu et al. (2024) and Hartmann et al. (2023) show that current language models have profound limitations when performing abstract reasoning, but Liu et al. (2023) indicate promising logical reasoning skills; however, performance is limited on out-of-distribution data. Multi-step reasoning is assessed by Chain-of-Thought Hub (Fu et al., 2023) and ThoughtSource (Ott et al., 2023), pointing out the limitations of language models in complex reasoning tasks.

Coverage of linguistic reasoning, which can be defined as the ability to understand and operate under the rules of language, has been limited in evaluation datasets for language models. One of the earliest examples is PuzzLing Machines (Şahin et al., 2020), which presents 7 different patterns from the Rosetta Stone paradigm Bozhanov and Derzhanski (2013) for models to perform exclusively machine translation. Chi et al. (2024) replicate Şahin et al. (2020)'s approach, manually creating a number of examples to avoid data leakage. Recently, some approaches have leveraged long context capabilities of language models to include in-context linguistic information (e.g. a grammar book (Tanzer et al., 2024) and other domain-specific sources (Zhang et al., 2024)) to solve different linguistic tasks. For large-scale linguistic reasoning evaluation, Big-Bench (Lewkowycz et al., 2022) includes a task linguistic mappings[3], relying on arbitrary artificial grammars to perform logical deduction. This approach is limited by its reliance on constructed languages instead of natural languages, which overlooks more complex underlying properties of languages, such as voicing rules. Finally, Waldis et al. (2024) present Holmes, a comprehensive benchmark for linguistic competence in English language.

## 3 BENCHMARKING LINGUISTIC REASONING

To overcome the previous limitations, we built a dataset where, in most cases, a model has no information about task language outside of the given context. To achieve this, we worked with problems extracted from the International Linguistic Olympiad.

### 3.1 IOL

The International Linguistic Olympiad (IOL)[4] is a contest for students up to secondary school level, where contestants must compete solving problems based on their understanding of linguistics (Derzhanski and Payne, 2010). The presented problems are formulated following the Rosetta Stone paradigm and present participants with challenges related to a variety of (mainly) extremely low-resource languages that students are not expected to be familiar with. The goal is for participants to leverage their linguistic skills rather than their foreign language knowledge. The IOL has been held yearly since 2003 (with the exception of 2020), and every year includes 5 short problems (to be solved individually) and 1 long, multipart problem (to be solved in groups). Problems are formulated in English and in several languages (up to 25 languages for the

---

[3]https://github.com/google/BIG-bench/blob/main/bigbench/benchmark_tasks/linguistic_mappings/
[4]https://ioling.org

2023 edition). The IOL corpus is available on their website in different formats of PDF with questions and correct answers, explanations of some answers and total marks for each problem. Beyond IOL, there are regional contests (e.g. Asia Pacific Linguistic Olympiad[5] and The Australian Computational and Linguistics Olympiad[6]) that award places for the IOL.

## 3.2 SELECTING PROBLEMS FOR OUR BENCHMARK

To select the types of questions for the dataset, we built a taxonomy exploring the IOL from 2003 to 2023. We excluded all instances for which their category only appears once; those where the question includes an image or those where the response is only an explanation. The remaining problems require solving different linguistic reasoning tasks, such as morphosyntactic segmentation (eg., verb conjugation), morphosemantic alignment (e.g., noun negation), derivation (e.g., finding cognates in related languages), morphophonological segmentation (e.g., pluralization) or graphophonemic transcription (e.g., transcription from one script to another). In total, Linguini is composed by 894 questions grouped in 160 problems across 75 (mostly) extremely low-resource language. A list of languages can be found in Appendix B. We classify the problems included in Linguini into the three categories according to their content: sequence transduction, fill-in-blanks and number transliteration. Figure 1 shows one example of each.

Figure 1: Examples of Linguini entries covering the three problems included in the dataset: sequence transduction, fill-in-blanks, number transliteration.

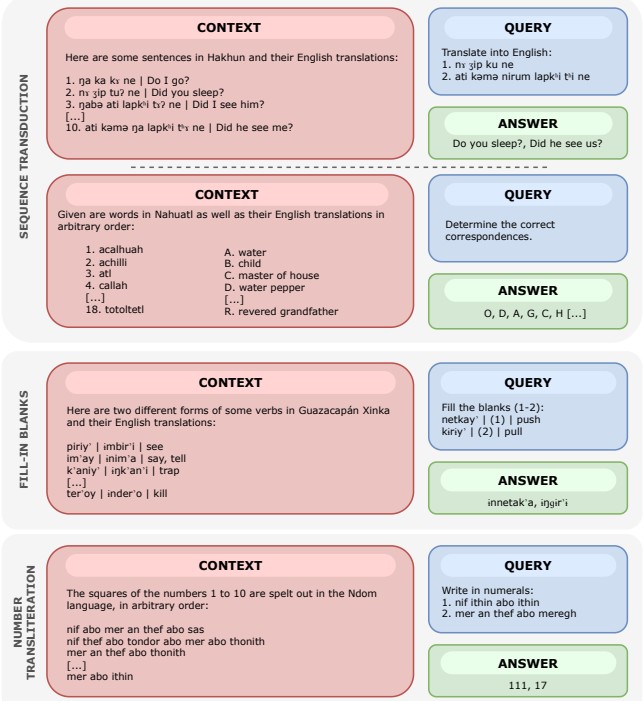

---

[5] https://aplo.asia
[6] https://ozclo.org.au

**Sequence transduction** This category includes sequence production (identified in the benchmark as `translation`) and sequence matching (identified as `match_letter`). The problems require the model to transform a sequence into a different space (e.g., language, phonetic representation, script) based on few examples. In some cases, basic phonetic/phonological knowledge is needed. For example, the model should be able to reason over principles of voicing and their implementation in situations of coarticulation. Some problems require to know that consonants come in voiced-voiceless pairs, and that one element of the pair may in some cases be a substitute for the other element in the pair under certain circumstances.

**Fill-in blanks** Fill-in blanks are mainly morphophonological derivation tasks, and they are identified in the benchmark as `fill_blanks`. Models need to understand what are the morphophonological rules that make it possible to go from the first form of a word to its second form. This can usually be applied to verbal (e.g., verb tense conjugation), nominal or adjectival (e.g., case declension) derivation. It involves understanding affixation rules and morpheme swapping rules, which often come with phonological rules if there are different coarticulation phenomena with different affixes or phonotactic phenomena such as consonantal mutations.

**Digit/text number transliteration** These problems are identified by the labels `text_to_num` and `num_to_text`. In them, models have to produce a digit or text equivalent, respectively. They require a model's understanding of morphological analysis and morpheme order.

Figure 2: A subset of the context of a problem in Terenâ language and the reasoning steps needed to solve it. To correctly answer the question, the model must notice that (a) voiced *d* mutates to voiceless paired sound *t* (fortition), (b) *n* is dropped because there are no voiceless nasal alveolar sounds and (c) an epenthetic vowel has to be added between the mutation consonant and the rest of the word (a root), and that the vowel that gets added matches the aperture of the vowel in the root. If the aperture is closed, the epenthetic vowel is the closed front vowel *i*; if the aperture is mid, the epenthetic vowel is the mid front vowel *e*.

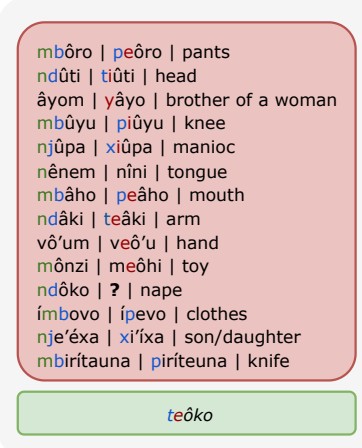

## 4 EXPERIMENTS

We perform zero-shot to few-shot (0-5 in-context examples) evaluation across the whole dataset for an array of open and proprietary LLMs. Given the size of the benchmark, we employ a leave-one-out cross-validation scheme to maximize the number of in-context candidates per task. For every given inference, we include examples of the same format (e.g., `translation`, `match_letter`), but we exclude in-context examples of the same language to avoid language contamination.

**Setup and Models**  We prompt models with an instruction, a context that provides information to unambiguously solve the linguistic problem and the problem itself. Scores of answers to each item of a problem are averaged to provide a single score (0-100) per task. We evaluate several major open LLMs and commercially available (behind API) SotA LLMs at the publication of this work. For open models, we conduct inference experiments in an 8 A100 GPUs node. An exhaustive list can be found in Appendix C.

**Evaluation**  We use exact match (accuracy) as main evaluation criterion. Given the almost null performance on exact match of certain models, we also include chrF (Popović, 2015) as a *softer* metric. A low chrF score indicates extremely low performance models, e.g. not understanding the domain of the task at hand.

## 5 RESULTS AND DISCUSSION

Table 1 shows there's a gap between the best performing open model and the best performing proprietary model, with several tiers of proprietary models above the best open model (*llama-3-70b*). We also find mixed impact of in-context examples in the performance of the models. While some models benefit from it (such as *llama-3-70b-it*), other models' performance degrades as the number of examples increases (such as *claude-3-opus*). This disparity might be due to the two factors introduced by the ICEs: from one side, they set an answer format that could be useful for models that can't infer it directly from a single natural language instruction and, from another side, they introduce tokens of languages potentially unrelated to the evaluated problem. It is possible that for models more capable of instruction following, only the second factor plays a role in the model's performance. We include results with chrF in Appendix E for reference.

Table 1: Exact match results with Linguini for 0-5 ICEs.

| Model | 0 | 1 | 2 | 3 | 4 | 5 | Best(↑) |
|---|---|---|---|---|---|---|---|
| claude-3-opus | 24.05 | 20.58 | 21.36 | 19.91 | 17.00 | 15.1 | 24.05 |
| gpt-4o | 14.65 | 12.98 | 13.87 | 12.98 | 13.98 | 13.76 | 14.65 |
| gpt-4 | 6.38 | 9.96 | 11.52 | 12.98 | 11.74 | 13.31 | 12.98 |
| claude-3-sonnet | 12.30 | 8.95 | 10.29 | 10.40 | 9.28 | 8.72 | 12.30 |
| gpt-4-turbo | 8.72 | 9.40 | 9.96 | 7.49 | 8.61 | 9.96 | 9.96 |
| llama-3-70b | 8.17 | 5.93 | 7.72 | 8.84 | 8.72 | 6.60 | 8.84 |
| llama-3-70b-it | 4.81 | 5.93 | 7.16 | 7.38 | 6.82 | 8.39 | 8.39 |
| claude-3-haiku | 6.04 | 7.61 | 4.36 | 6.04 | 6.94 | 7.05 | 7.61 |
| llama-2-70b | 4.70 | 2.24 | 2.57 | 3.24 | 3.36 | 3.58 | 3.58 |
| mistral-0.1-8x7b | 2.46 | 3.47 | 3.91 | 3.02 | 3.24 | 3.47 | 3.91 |
| llama-2-70b-it | 0.89 | 1.45 | 2.80 | 3.02 | 3.13 | 2.80 | 3.13 |
| gemma-2b | 0.34 | 2.01 | 1.90 | 1.34 | 1.45 | 1.90 | 2.01 |
| qwen-1.5-110b-it | 1.45 | 1.23 | 1.34 | 1.45 | 1.45 | 1.68 | 1.68 |

In addition to our main experiments, we performed a series of ablation studies to get a better insight of how language models perform linguistic reasoning.

## 5.1 NO-CONTEXT PROMPTING

Given that we don't have information about training data for the majority of the analyzed models, we performed a series of experiments to study the degree in which models rely on the given context to provide correct answers. Models that have not been trained on any data of the task language should have a null-adjacent performance when not given the context necessary to solve the task. We analyze the impact of ignoring the context provided in the benchmark as a proxy of possible data contamination. The results are shown in Table 2.

Table 2: No context results

| Model | Zero-shot | No context | Δ |
|---|---|---|---|
| llama-3-70b-it | 4.81 | 1.12 | -3.69 |
| gpt-4-turbo | 8.72 | 1.45 | -7.27 |
| gpt-4 | 6.38 | 1.34 | -5.04 |
| claude-3-sonnet | 12.30 | 2.01 | -10.29 |
| mistral-0.1-8x7b | 2.46 | 1.98 | -0.48 |
| claude-3-haiku | 6.04 | 1.12 | -4.92 |
| qwen-1.5-110b-it | 1.45 | 0.43 | -1.02 |
| gemma-2b | 0.34 | 0.09 | -0.25 |
| llama-2-70b | 4.70 | 1.07 | -3.63 |
| llama-2-70b-it | 0.89 | 0.56 | -0.33 |
| llama-3-70b | 8.17 | 1.67 | -6.50 |
| claude-3-opus | 24.05 | 1.23 | -22.82 |
| gpt-4o | 14.65 | 1.45 | -13.20 |

We find steep performance drops for every model, which points towards a low likelihood of the language (or the training examples) being present in the models' training sets.

## 5.2 CHARACTER-WISE SUBSTITUTION

Since most problems are presented in Latin script, we wanted to understand whether the script in which the task languages are presented impact the performance on Linguini. But given that all information needed to solve the task is present in the context, the script should not have a major impact on the performance beyond encoding constraints. In other words, if the model doesn't rely on instances of the language (or the problem) in its training set, it should be able to solve the task in a non-Latin script as well. We selected the best performing model (*claude-3-opus*) and transcribed the best performing problems (those with accuracy greater than or equal to 75%) into 4 non-Latin alphabetical scripts (Cyrilic, Greek, Georgian and Armenian)[7]. An example of a transliterated problem can be found in Figure 3. Given the difficulty of uniformly transcribing a diverse set of orthographic systems and diacritics, we opted for performing a character/bi-character-wise substitution of the standard Latin alphabet character, leaving non-standard characters with their original Unicode symbol. We filtered 17 well performing problems, and excluded one with a non-Latin script task language (English Braille). We performed transcriptions on the remaining 16 problems.

Table 3 shows that the model retains the capacity to perform linguistic reasoning even after changing scripts, which backs the hypothesis of the model relying mainly on the presented context and not on spurious previous knowledge. The fact that for 13 our of 16 of the given problems there's at least one non-Latin

---

[7]The mappings from Latin script to the rest can be found at https://github.com/barseghyanartur/transliterate/

Figure 3: Example of transliteration of a problem into Cyrillic, Greek, Georgian and Armenian scripts.

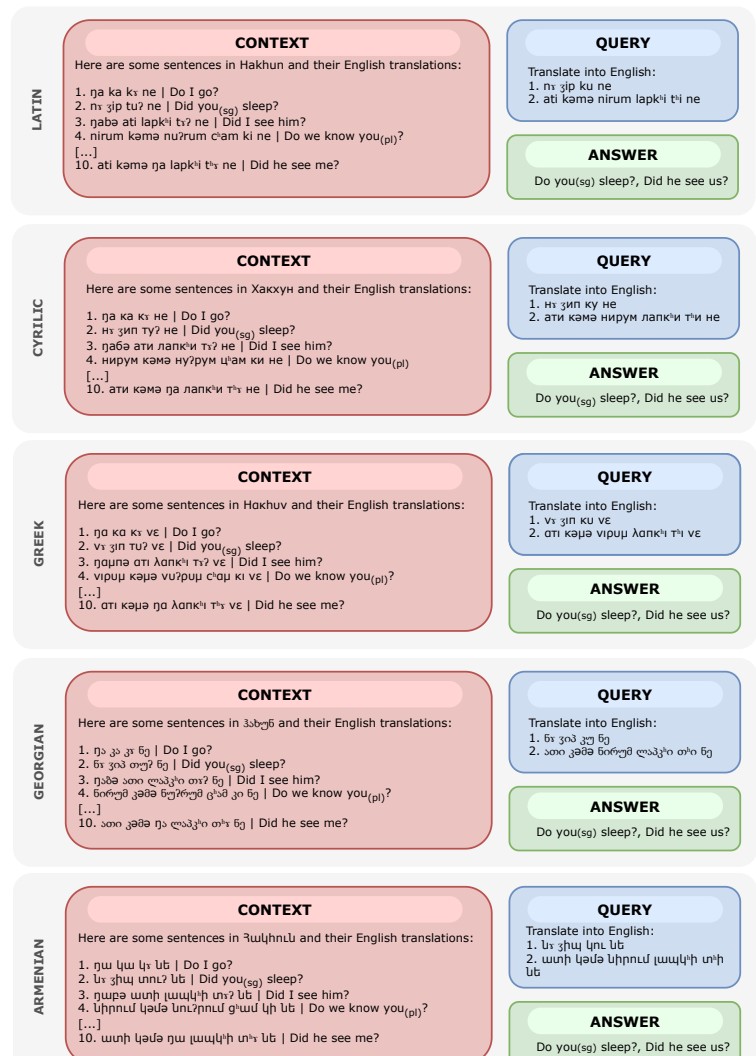

script in which the model can solve the problem with greater or equal performance than with Latin script further supports this claim. Performance disparity among scripts could be related to either the difference in tokenization of different scripts or to the inherent limitations of our transliteration strategy (e.g. the Armenian script might lack a specific consonant cluster that needs to be developed to provide the right answer, and character/bi-character-wise substitution doesn't take this nuance into account).

Table 3: Scores of selected problems with different language scripts for *claude-3-opus*.

| Problem code & language | Latn | Cyrl | Grek | Geor | Armn |
|---|---|---|---|---|---|
| 012023010100 (qda-gua) | 75.00 | 100.00 | 75.00 | 100.00 | 0.00 |
| 012021020500 (zun) | 100.00 | 0.00 | 100.00 | 0.00 | 0.00 |
| 012012030100 (eus) | 78.57 | 7.14 | 92.86 | 0.00 | 0.00 |
| 012018020100 (nst-hkn) | 83.33 | 83.33 | 66.67 | 83.33 | 100.00 |
| 012007050100 (tur) | 75.00 | 75.00 | 50.00 | 37.50 | 50.00 |
| 012006020100 (cat) | 75.00 | 50.00 | 50.00 | 58.33 | 33.33 |
| 012003030200 (eus) | 100.00 | 100.00 | 75.00 | 100.00 | 100.00 |
| 012004010100 (txu) | 100.00 | 100.00 | 66.67 | 66.67 | 33.33 |
| 012007030100 (kat) | 80.00 | 13.33 | 6.67 | 100.00 | 0.00 |
| 012009050100 (nci) | 83.33 | 83.33 | 83.33 | 83.33 | 50.00 |
| 012015020100 (kbd-bes) | 100.00 | 66.67 | 100.00 | 66.67 | 83.33 |
| 012012050100 (rtm) | 100.00 | 100.00 | 100.00 | 100.00 | 100.00 |
| 012011040200 (nci) | 100.00 | 50.00 | 75.00 | 75.00 | 0.00 |
| 012013010200 (yii) | 100.00 | 100.00 | 100.00 | 75.00 | 100.00 |
| 012012030200 (eus) | 100.00 | 50.00 | 0.00 | 0.00 | 0.00 |
| 012012030300 (eus) | 100.00 | 50.00 | 100.00 | 0.00 | 0.00 |
| Average | 85.71 | 56.12 | 65.31 | 63.27 | 38.78 |

## 5.3 LANGUAGE RESOURCEFULNESS AND ACCURACY

We were also interested in assessing whether higher-resource languages perform, on average, better than lower-resource languages. We use two metrics as proxies of language resourcefulness: number of speakers (Figure 4) and online presence (Figure 5), measured by Google searches.

Figure 4: Accuracy vs. number of speakers. Data points are clustered for readability.

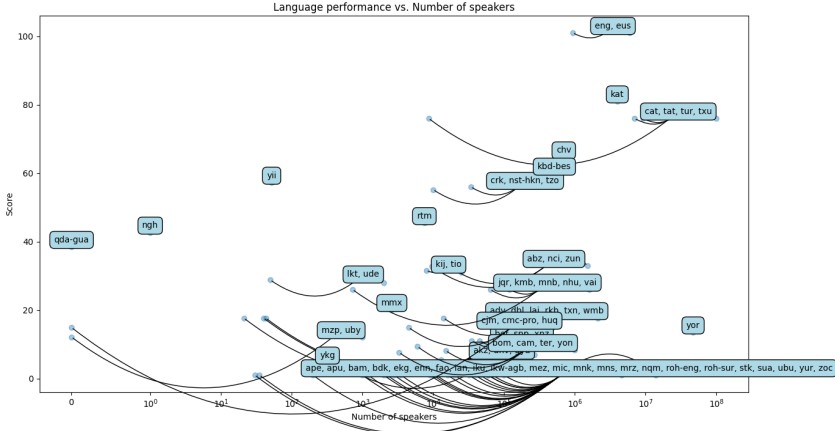

We find the distribution to follow a uniform trend with respect to both metrics of language resourcefulness, which suggests that the accuracy isn't largely correlated to to its likelihood of being included in the training

Figure 5: Accuracy vs. number of Google searches. Data points are clustered for readability.

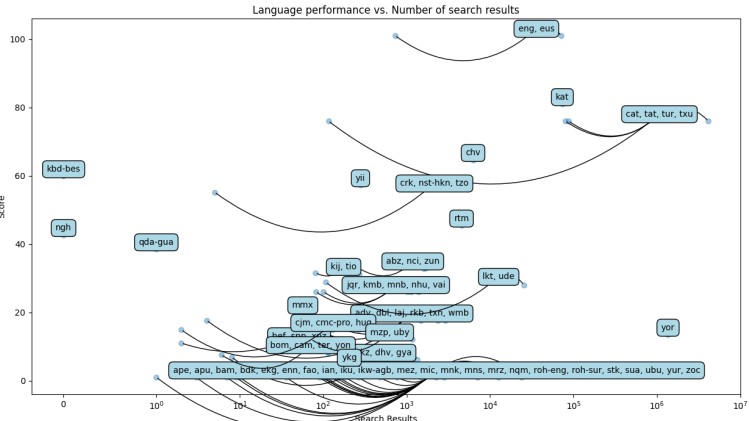

set. Notable exceptions to this trend are a number of very high-resource languages (e.g., cat, eus, kat, tur), which are very likely to be included in the model's training set, given their institutional status.

### 5.4 ONE-BOOK PROMPTING

Previous studies (Tanzer et al., 2024) have shown the capacity of language models to acquire some proficiency in the task of machine translation for an unseen language only through an in-context textbook. We leverage publicly available textbooks to scale Tanzer et al. (2024)'s analysis in number of languages and types of tasks. We convert the textbooks in PDF format to raw text using the pdftotext library[8] and include them as context without any pre-processing. A list of textbooks employed can be found in Appendix D.

Table 4: Scores for a subset of examples evaluated with no context, with context, with a textbook and with a combination of both

| Language code | No-context | Context | Textbook | Context + Textbook |
|---|---|---|---|---|
| akz | 0.00 | 5.13 | 0.00 | 3.85 |
| apu | 0.00 | 0.00 | 0.00 | 16.67 |
| mnk | 0.00 | 0.00 | 0.00 | 0.00 |
| Average | 0.00 | 1.71 | 0.00 | 6.84 |

Even thought in many cases the orthography of the task language greatly varies from the textbook to the problem and the PDF to text conversion introduces errors for highly diacritical text (as shown in Figure 6), the results in Table 4 show that a model can learn to model linguistic phenomena relying on a single in-context textbook.

---

[8]https://github.com/jalan/pdftotext

Figure 6: Example of transliteration of a problem into Cyrillic, Greek, Georgian and Armenian scripts. The discrepancies between the term *kyky* (English: *man*) in the original document (a scan from a 1894 grammar book of Apurinã language), its OCR conversion and the text of a problem in the benchmark are highlighted. In spite of the noise introduced by different orthographies and imperfect OCR, performance for Apurinã increases from 0% 16.67% with the full OCR text in-context.

## 6 CONCLUSIONS

We presented Linguini, a new linguistic reasoning evaluation dataset. Our experiments show that Linguini provides a compact and effective benchmark to assess linguistic reasoning without relying on a substrate of existing language-specific knowledge. There's a considerable gap between open source and proprietary LLMs in linguistic reasoning. Subsequent experiments also show very low likelihood of dataset contamination in the analyzed models. Limitations and broader impact of the dataset are discussed in Appendix A.

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

## A   LIMITATIONS, FURTHER WORK AND BROADER IMPACT

Evaluation of long in-context learning for linguistic reasoning is limited in this paper to a few languages, given the difficulties of finding publicly available grammar books. We plan to scale up the number of covered languages in further versions of the benchmark to perform a better encompassing analysis of long in-context learning.

Our dataset also lacks a curated list of explanations for each problem, which could be used as a basis to run chain-of-thought experiments and improve lingusitic reasoning skills of language models. We intend to engage with linguists and IOL organizers to fill this gap.

This benchmark intends to address and quantify the root of multilingualism, which in turn can impact the support of language models for the majority of world languages.

## B   LANGUAGES OF LINGUINI

Table 5: Languages and their characteristics

| Lang. Code | Language | No. Speakers[9] | No. Search Results[10] | Language Family | Script |
|---|---|---|---|---|---|
| abz | Abui | 16,000 | 263 | Trans-New Guinea | Latin |
| ady | Adyghe | 425,000 | 2,370 | Abkhaz-Adyghe | Latin |
| akz | Alabama | 370 | 1,350 | Muskogean | Latin |
| abz | Mountain Arapesh | 16,000 | 98 | Torricelli | Latin |
| apu | Apurinã | 2800 | 264 | Maipurean | Latin |
| bam | Bambara | 14000000 | 7150 | Niger-Congo | N'Ko |
| bdk | Budukh | 200 | 126 | Nakh-Daghestanian | Latin |
| bef | Bena Bena | 45000 | 107 | Trans-New Guinea | Latin |
| bom | Birom | 1000000 | 115 | Niger-Congo | Latin |
| cam | Cemuhî | 3300 | 6 | Austronesian | Latin |
| cat | Catalan | 9200000 | 87100 | Indo-European | Latin |
| chv | Chuvash | 700000 | 6260 | Turkic | Latin |
| cjm | Phan Rang Cham | 491448 | 2 | Austronesian | Latin |
| cmc-pro[11] | Proto-Chamic | 0 | 267 | Austronesian | Latin |
| crk | Plains Cree | 34000 | 5290 | Algic | Latin |
| dbl | Dyirbal | 21 | 2900 | Australian | Latin |
| dhv | Drehu | 13,000 | 216 | Austronesian | Latin |
| ekg | Ekari | 100000 | 141 | Trans-New Guinea | Latin |
| eng | English Braille | 6000000 | 728 | Indo-European | Latin |
| enn | Engenni | 20000 | 185 | Niger-Congo | Latin |
| eus | Basque | 936,812 | 71100 | Isolate | Latin |
| fao | Faroese | 69000 | 23800 | Indo-European | Latin |
| gya | Northwest Gbaya | 267000 | 8 | - | Latin |
| huq | Tsat | 4500 | 128 | Austronesian | Latin |
| ian | Iatmül | 46000 | 9 | Papua New Guinea | Latin |
| iku | Inuktitut | 39,000 | 12500 | Eskimo-Aleut | Latin |
| ikw-agb[11] | Agbirigba | 30 | 1 | Niger-Congo | Latin |
| jqr | Jaqaru | 725 | 101 | Aymaran | Latin |
| kat | Georgian | 4000000 | 73700 | Kartvelian | Latin |
| kbd-bes[11] | Besleney Kabardian | 516000 | 0 | Abkhaz-Adyghe | Latin |
| kij | Kilivila | 25000 | 271 | Austronesian | Latin |
| kmb | Kimbundu | 1600000 | 1130 | Niger-Congo | Latin |
| laj | Lango | 2100000 | 1490 | Nilo-Saharan | Latin |
| lkt | Lakhota | 2000 | 25300 | Siouan-Catawban | Latin |
| mez | Menominee | 2000 | 2240 | Algic | Latin |
| mic | Micmac | 11000 | 774 | Algic | Latin |
| mmx | Madak | 2600 | 57 | Austronesian | Latin |
| mnb | Muna | 270000 | 1020 | Austronesian | Latin |
| mnk | Maninka | 4600000 | 478 | Niger-Congo | N'Ko |
| mns | Mansi | 2229 | 1490 | Uralic | Latin |
| mrz | Coastal Marind | 9000 | 100 | Trans-New Guinea | Latin |
| mzp | Movima | 1000 | 72 | Isolate | Latin |
| nci | Classical Nahuatl | 1500000 | 1690 | Uto-Aztecan | Latin |
| ngh | Nluuki | 1 | 0 | Tuu | Latin |
| nhu | Nooni | 64000 | 82 | Niger-Congo | Latin |
| nqm | Ndom | 1200 | 154 | Trans-New Guinea | Latin |
| nst-hkn[11] | Hakhun | 10000 | 5 | Sino-Tibetan | Latin |
| qda-gua[11] | Guazacapán Xinka | 0 | 1 | Xincan | Latin |
| rkb | Rikbaktsa | 40 | 54 | Isolate | Latin |

| Lang. Code | Language | No. Speakers | No. Search Results | Language Family | Script |
|---|---|---|---|---|---|
| roh-eng[10] | Engadine | 60000 | 7 | Indo-European | Latin |
| roh-sur[11] | Sursilvan | 60000 | 3 | Indo-European | Latin |
| rtm | Rotuman | 7500 | 4560 | Austronesian | Latin |
| spp | Supyire | 460000 | 45 | Niger-Congo | Latin |
| stk | Arammba | 1000 | 36 | South-Central Papuan | Latin |
| sua | Sulka | 3500 | 107 | Isolate | Latin |
| tat | Tatar | 7000000 | 79700 | Turkic | Latin |
| ter | Terêna | 15,000 | 115 | Maipurean | Latin |
| tio | Teop | 8000 | 81 | Austronesian | Latin |
| tur | Turkish | 100000000 | 4130000 | Turkic | Latin |
| txn | West Tarangan | 14,000 | 4 | Austronesian | Latin |
| txu | Kayapo | 8600 | 116 | Jean | Latin |
| tzo | Tzotzil | 550000 | 1160 | Mayan | Latin |
| ubu | Umbu-Ungu | 32,000 | 90 | Trans-New Guinea | Latin |
| uby | Ubykh | 0 | 1180 | Abkhaz-Adyghe | Latin |
| ude | Udihe | 50 | 108 | Tungusic | Latin |
| vai | Vai | 120000 | 1380 | Niger-Congo | Latin |
| wmb | Wambaya | 43 | 112 | Australian | Latin |
| xnz | Kunuz Nubian | 35000 | 2 | Nilo-Saharan | Latin |
| yii | Yidiny | 52 | 280 | Australian | Latin |
| ykg | Tundra Yukaghir | 320 | 206 | Yukaghir | Latin |
| yon | Yonggom | 6,000 | 48 | Trans-New Guinea | Latin |
| yor | Yoruba | 47000000 | 1360000 | Niger-Congo | Latin |
| yur | Yurok | 35 | 2830 | Algic | Latin |
| zoc | Copainalá Zoque | 10000 | 10 | Mixe-Zoquean | Latin |
| zun | Zuni | 9500 | 1610 | Isolate | Latin |

## C MODELS

Table 6: Overview of Large Language Models

| Model ID | API Version | Organization | Model Size[12] | Open | Reference |
|---|---|---|---|---|---|
| claude-3-opus | claude-3-opus-20240229 | Anthropic | - | ✗ | Anthropic AI (2024) |
| gpt-4o | gpt-4o-2024-05-13 | OpenAI | - | ✗ | OpenAI (2024) |
| gpt-4 | gpt-4-0125-preview | OpenAI | - | ✗ | OpenAI (2024) |
| claude-3-sonnet | claude-3-sonnet-20240229 | Anthropic | - | ✗ | Anthropic AI (2024) |
| gpt-4-turbo | gpt-4-turbo-2024-04-09 | OpenAI | - | ✗ | OpenAI (2024) |
| llama-3-70b | - | Meta | 70.6 | ✓ | AI@Meta (2024) |
| llama-3-70b-it | - | Meta | 70.6 | ✓ | AI@Meta (2024) |
| claude-3-haiku | claude-3-haiku-20240307 | Anthropic | - | ✗ | Anthropic AI (2024) |
| llama-2-70b | - | Meta | 69.0 | ✓ | Touvron et al. (2023) |
| mistral-0.1-8x7b | - | Mistral | 46.7 | ✓ | Jiang et al. (2024) |
| llama-2-70b-it | - | Meta | 69.0 | ✓ | Touvron et al. (2023) |
| gemma-2b | - | Google | 2.5 | ✓ | Gemma Team (2024) |
| qwen-1.5-110b-it | - | Alibaba | 111.0 | ✓ | Bai et al. (2023) |

## D BOOKS

## E CHRF RESULTS

---

[9] According to Eberhard et al. (2020)

[10] Number of search results of the exact string "<Language name> language" using Google Seach API

[11] Language code not in ISO-639-3

[12] in billion parameter

Table 7: Overview of Grammar Books [tba]

| Language | Book Title | Citation |
|----------|-----------|----------|
| akz | The Language of the Alabama Indians | Lupardus (1982) |
| apu | A Grammar and a Vocabulary of the Ipuriná Language | Polak (1894) |
| mnk | The Structure of Faranah-Maninka | Spears (1965) |

Table 8: chrF results with Linguini for 0-5 ICEs

| Model | 0 | 1 | 2 | 3 | 4 | 5 |
|-------|---|---|---|---|---|---|
| llama-3-70b-it | 45.35 | 42.65 | 43.89 | 45.99 | 48.07 | 51.08 |
| gpt-4-turbo | 52.89 | 50.82 | 50.03 | 50.94 | 49.98 | 51.79 |
| gpt-4 | 44.62 | 55.05 | 58.47 | 57.36 | 57.62 | 58.18 |
| claude-3-sonnet | 54.97 | 45.32 | 50.91 | 47.35 | 46.51 | 42.06 |
| mistral-0.1-8x7b | 42.0 | 34.8 | 38.01 | 37.57 | 37.64 | 37.63 |
| claude-3-haiku | 47.74 | 50.75 | 41.02 | 45.38 | 42.32 | 41.83 |
| qwen-1.5-110b-it | 2.57 | 0.0 | 0.22 | 0.78 | 1.12 | 2.8 |
| gemma-2b | 33.72 | 27.19 | 24.62 | 26.04 | 27.04 | 27.63 |
| llama-2-70b | 45.3 | 35.39 | 34.06 | 35.54 | 36.21 | 36.44 |
| llama-2-70b-it | 43.55 | 41.42 | 39.73 | 41.42 | 39.69 | 39.34 |
| llama-3-70b | 37.25 | 36.04 | 41.83 | 41.21 | 41.92 | 41.63 |
| claude-3-opus | 63.96 | 58.26 | 58.5 | 53.17 | 49.01 | 46.55 |
| gpt-4o | 57.68 | 58.13 | 57.32 | 58.86 | 58.99 | 58.22 |

