# OpenReview forum: "Linguini: A benchmark for language-agnostic linguistic reasoning"
_ICLR.cc/2025/Conference — Submitted to ICLR 2025_

### Official Review · Reviewer_Bs42 · 2024-11-02

**Soundness:** 2
**Presentation:** 3
**Contribution:** 2
**Rating:** 5
**Confidence:** 4

**Summary:**

This paper introduces Linguini, a new benchmark for evaluating language models' linguistic reasoning skills.
The benchmark is based on problems extracted from the International Linguistic Olympiad (IOL), which is designed to test meta-linguistic awareness and deductive reasoning abilities.
The authors demonstrate that Linguini provides a more challenging and nuanced evaluation of linguistic reasoning skills compared to existing benchmarks, particularly when considering models' ability to generalize across languages.

Overall, I think this is a clearly written and well-motivated benchmark; however, the analysis could be more detailed, and the dataset construction process could be more transparent.
More discussion on why the benchmark is realistic and how it affects real-world applications would also be helpful.

**Strengths:**

- This is a creative benchmark to evaluate complex reasoning skills in language models.
- The benchmark is based on a diverse set of problems extracted from the International Linguistic Olympiad, covering multiple languages and linguistic phenomena.
- The dataset construction process is mostly clear.

**Weaknesses:**

The paper is mostly clear, and the benchmark is well-motivated, but there are a few areas that could be improved:
- As the authors have discussed, the sampled set of problems is possibly biased towards phonographic languages, where the logographic languages, for which transliteration does lose some information, are not well represented. It would be interesting to see how vision-language models perform on these problems using screenshots as the input.
- While we can indeed call IOL problems require "linguistic reasoning" skills, these skills are essentially a form, or a combination of inductive, deductive, and abductive reasoning skills. It would be interesting to ground each of the problems to the required skills and perform a more in-depth analysis. The current analysis in this paper seems to be more on a coarse level.
- As will be raised in my first question, it's not clear how the authors handled diacritical marks in the original IOL problems. This is an important issue that could affect the quality of the benchmark, and this might affect the one-book prompting performance of the models.
- It is unclear to me whether the benchmark is realistic and how it affects real-world applications. It would be helpful to discuss this in more detail.

**Questions:**

The most important question on my side is how the authors converted the IOL original problems (in PDF format; as mentioned in Section 3.1) to textual format that language models can accept.
1. Specifically, many, if not most, of the IOL problems have diacritical marks on some letters (e.g, ô, õ, and ç)
To the best of my knowledge, current pdf-to-text converters do not handle these marks very well---even directly copying the text from a PDF file and pasting it into a text editor can result in missing or incorrect diacritical marks, especially if the problem set is written in LaTeX.
I noticed that the authors might have briefly mentioned the issue in Section 5.2, but it's not very clear to me.
Can the authors provide more details on how they handled this issue?

Below are some more detailed questions/comments:
2. What do the lines in Figures 4 and 5 represent?
3. Following question 1, it would be interesting to see the performance of models vs. the presence of the language in the training data.

**Details Of Ethics Concerns:**

While this looks fine to me, I would defer to the ethics committee to determine whether the benchmark collection process has possible copyright issues, as all the problems are collected from the International Linguistic Olympiad.

---

### Official Review · Reviewer_esg2 · 2024-11-03

**Soundness:** 4
**Presentation:** 3
**Contribution:** 2
**Rating:** 6
**Confidence:** 3

**Summary:**

The paper presents Linguini, a novel benchmark to assess linguistic capabilities. The questions in the benchmark test linguistic reasoning, which refers to the ability to understand lesser known language from information provided in the context *only* and answer questions based on that understanding. The work ensures that to answer the question, no prior knowledge of the language is required (the authors perform a variety of experiments to verify there is no contamination). The benchmark is tested across various open source, as well proprietary LLMs. The authors find a significant gap between open and proprietary LLMs, as well as other interesting findings.

**Strengths:**

1. The introduced benchmark is fairly challenging, given the SoTA performance is below 25%.
2. The experiments are very scientifically conducted and thorough. For example, “no context prompting” experiment in 5.1 showed evidence of lack of presence of language data in models training. Another example is 5.3, which shows that unless a language is higher on the resource scale, scores remain low.
3. The paper is well written, and makes for an interesting read.

**Weaknesses:**

1. The goal of this work is to create a benchmark to evaluate linguistic skills of the model (unrelated to language specific learning). It would be good to fully understand why this is an important problem? Can a toy dataset be built instead, something that tests linguistic abilities, but isn’t a real language?
2. The paper is well written, but it would be good to improve some areas, such as:
- The related work could use more detail. For instance, it’d be important to add information about very low-resource languages and related attempts at benchmarking.
- Some specific stats on how Linguini was constructed should be included. The dataset includes 894 examples with some filtering. How many were filtered out from IOL?
- For a non-linguistic audience, sec 3.2 should include more details and references. Terms used such as: “verb tense conjugation”, “case declension”, “principles of voicing” need some explanations.
- Section 5 results indicate that for some models, performance degrades as in context examples are increased? This is counterintuitive. How were the examples selected?

**Questions:**

See weakness section. I'd particularly like a clear answer for 1.

---

### Official Review · Reviewer_G7i7 · 2024-11-04

**Soundness:** 2
**Presentation:** 3
**Contribution:** 2
**Rating:** 5
**Confidence:** 3

**Summary:**

The paper proposes a new benchmark, Linguini, to test the inference abilities of language models based on fundamental linguistic knowledge, independent of pre-training language familiarity. Linguini focuses on 75 low-resource languages sourced from the International Linguistic Olympiad corpus and includes three types of tasks: sequence transduction, fill-in-blanks, and number transliteration. Experiments are designed with few-shot prompting, where models are expected to acquire all necessary knowledge about the language within the provided context. Tests conducted on 13 different open and closed LLMs revealed low performance, suggesting that Linguini presents a significant challenge for current language models.

**Strengths:**

1. The experiments were conducted using a diverse set of 13 open and closed LLMs, enhancing the generalizability of the conclusions.
2. The selection of low-resource languages unlikely to appear in pre-training data and the dataset design that focuses on pure linguistic reasoning abilities are interesting and present a novel idea.
3. The benchmark was rigorously tested from various perspectives, providing a high-quality resource for the community:
    - Experiments without context indicate that there is no data leakage.
    - Using non-Latin scripts helped verify that the models are not relying on Latin-script language cues from the training data for inference, ruling out such reliance.
    - Examining the relationship between language resourcefulness (proxied by the number of speakers and Google search results) and model performance showed that higher-resource languages tended to yield better performance.
    - Adding explanatory texts (textbooks) for each language improved performance, indicating that language models can learn linguistic abilities from explanatory content.

**Weaknesses:**

1. It is unclear how low-resource the selected 75 languages are for the language models used in the experiments. Although it is likely that they are low-resource, this assumption alone does not ensure reliability in the experimental results. Since, for some of the models, pre-training data is publicly available, verifying the presence of these languages in the pre-training data would strengthen the paper.
2. The motivation for putting the constraint that models must learn and utilize the characteristics of previously unseen languages solely through in-context learning is unclear. Language models typically acquire linguistic knowledge through next-token prediction during pre-training, using this knowledge in inference to make few-shot prompting (sometimes with finetuning) effective. However, this study assumes models need to learn language characteristics from context alone for languages not encountered in pre-training. While I understand that the proposed benchmark aims to test only language model capabilities, the broader impact of this study on future LLM research is somewhat unclear due to the huge difference of these assumptions.
3. Typos
    - L.417: “Even thought” => “Even though”

**Questions:**

1. For Weakness 1, is it possible to check the language distribution in the training data of some language models? If so, I would like to see that distribution.
2. For Weakness 2, what is the motivation behind the constraint that models must learn and utilize linguistic characteristics solely through in-context learning? How might this finding contribute to future LLM research?

---

### Official Review · Reviewer_mh2p · 2024-11-07

**Soundness:** 4
**Presentation:** 2
**Contribution:** 1
**Rating:** 3
**Confidence:** 4

**Summary:**

This paper proposes Linguini, a new benchmark to evaluate the linguistic reasoning skills of LLMs. The authors describe the creation of the benchmark and apply it to several open and closed LLMs. They report a pronounced gap in performance between these two classes of models and conduct several post-hoc analyses.

**Strengths:**

Linguistic reasoning is an exciting way to analyze the capabilities of LLMs, combining several important skills (e.g., reasoning, language skills). The creation of the benchmark is well-motivated and sound. The authors evaluated a large number of LLMs, with some interesting findings (e.g., the difference between open and closed models, the results of the character substitution experiment).

**Weaknesses:**

The key weakness of the paper is that there already exists a benchmark for testing the linguistic reasoning skills of LLMs, specifically LINGOLY ([Bean et al., 2024](https://arxiv.org/abs/2406.06196)), also comprised of puzzles from the Linguistic Olympiad. It seems that LINGOLY has a wider scope than Linguini, so it is not clear what Linguini adds beyond LINGOLY. The authors also do not discuss the difference between their benchmark and LINGOLY -- in fact, they do not even mention LINGOLY. (I assume that the authors started working on Linguini at a time when LINGOLY had not been released yet, but by the ICLR submission deadline, four months had passed since its release.)

**Questions:**

Minor points:
- Section 5.1: This analysis could be more in-depth. Was there a pattern in terms of which languages showed a smaller gap across LLMs? It would be particularly interesting to see commonalities between LLMs trained on similar data (e.g., models from the same model family).
- Section 5.2: You mention the potential impact of the tokenizer. I agree with this hypothesis -- given the fact that tokenizations for low-resource scripts have a higher fragmentation rate ([Ahia et al., 2023](https://aclanthology.org/2023.emnlp-main.614/)), it is likely that the transliteration results in tokenizations that are close to sequences of individual characters. Given the problems that subword-level tokenization (as opposed to character-level tokenization) causes for linguistic and particularly morphological tasks ([Hofmann et al., 2021](https://aclanthology.org/2021.acl-long.279/)), this would explain why the LLMs are sometimes better at solving the linguistic problems included in Linguini after the transliteration. I would love to see a more in-depth analysis, especially since you still have some space left.

Typos:
- 38: "needs" -> "needed"
- 107: "language" -> "languages"
- 375: "to to its likelihood" -> "to its likelihood"

---

### Meta-Review · Area_Chair_a6xA · 2025-01-03

**Metareview:**

This paper introduces Linguini, a benchmark for evaluating language models' linguistic reasoning capabilities using problems from the International Linguistic Olympiad (IOL). The benchmark is comprised by 894 questions spread across 75 low-resource languages, designed to test the ability to reason about linguistic patterns without relying on pre-existing language knowledge. The authors evaluate 13 different language models and find that performance is generally poor (<25% accuracy), with a notable gap between closed and open-source models (24% vs 9%).

The benchmark addresses an important aspect of language model evaluation - the ability to reason about linguistic patterns independently of training data exposure. The experimental design is rigorous, including controls for data contamination, as well as some instructive experiments (e.g., testing the effects of adding textbooks for each language to check if models can learn linguistic abilities from explanatory content).

The most significant weakness is the paper's relationship to LINGOLY (Bean et al., 2024), an existing benchmark also based on IOL problems, which covers more languages and contains more examples. Although published very recently, it can not be considered contemporaneous, since it was released 4 months before the ICLR submission deadline. The current submission does not acknowledge or differentiate itself from this prior work.

Given the fundamental issue of overlap with LINGOLY, I recommend rejection while encouraging the authors to revise and resubmit with clearer differentiation from prior work and more in-depth technical analysis as suggested by the reviewers.

**Additional Comments On Reviewer Discussion:**

Reviewer mh2p brought up the main issue with the paper, i.e., lack of discussion of how this work is differentiated from Bean et al., 2024. Reviewers also requested clarification of a few technical points, including filtering methods used in example selection, more in-depth analysis of required reasoning skills, and a discussion of whether similar skills could be tested with, e.g., constructed artificial data. No author rebuttal was provided during the discussion period.

---

### Decision · Program_Chairs · 2025-01-22

Reject